# Membrane contact sites regulate vacuolar fission via sphingolipid metabolism

Kazuki Hanaoka[1†], Kensuke Nishikawa[1†], Atsuko Ikeda[1†], Philipp Schlarmann[1], Saku Sasaki[1], Sotaro Fujii[1], Sayumi Yamashita[2], Aya Nakaji[2], Kouichi Funato[1,2]*

[1]Graduate School of Integrated Sciences for Life, Hiroshima University, Higashi-Hiroshima, Japan; [2]School of Applied Biological Science, Hiroshima University, Higashi-Hiroshima, Japan

*For correspondence:
kfunato@hiroshima-u.ac.jp

[†]These authors contributed equally to this work

Competing interest: The authors declare that no competing interests exist.

**Abstract** Membrane contact sites (MCSs) are junctures that perform important roles including coordinating lipid metabolism. Previous studies have indicated that vacuolar fission/fusion processes are coupled with modifications in the membrane lipid composition. However, it has been still unclear whether MCS-mediated lipid metabolism controls the vacuolar morphology. Here, we report that deletion of tricalbins (Tcb1, Tcb2, and Tcb3), tethering proteins at endoplasmic reticulum (ER)–plasma membrane (PM) and ER–Golgi contact sites, alters fusion/fission dynamics and causes vacuolar fragmentation in the yeast *Saccharomyces cerevisiae*. In addition, we show that the sphingolipid precursor phytosphingosine (PHS) accumulates in tricalbin-deleted cells, triggering the vacuolar division. Detachment of the nucleus–vacuole junction (NVJ), an important contact site between the vacuole and the perinuclear ER, restored vacuolar morphology in both cells subjected to high exogenous PHS and Tcb3-deleted cells, supporting that PHS transport across the NVJ induces vacuole division. Thus, our results suggest that vacuolar morphology is maintained by MCSs through the metabolism of sphingolipids.

## eLife assessment

This manuscript presents **valuable** findings that contribute to our understanding of how sphingolipids and membrane contact sites, formed by the tethering protein family tricalbins, are involved in regulating vacuolar morphology in *S. cerevisiae*. The evidence supporting the authors' claims is largely **solid**. While the reported correlation between sphingolipid levels and vacuole homeostasis is interesting and intriguing, more work is needed to thoroughly substantiate the proposed mechanism. This study will be of interest to cell biologists focusing on intracellular organization and lipid metabolism.

## Introduction

Cellular organelles are formed by membranes with unique lipid compositions, morphology, and functions. The vacuole of budding yeast is an organelle that shares many similarities with mammalian lysosomes and plant vacuoles. Vacuoles possess degradative and storage capacities, and are essential for maintaining cellular homeostasis, including maintaining pH or ion homeostasis, cellular detoxification, and responses to osmotic shock and nutrient environments. Vacuoles are liable to change their morphology (size, volume, and number) through cycles of fusion and fission, to adapt to the intra- and extracellular environments or upon vacuole inheritance to daughter cells. The steady-state morphology of the vacuole is maintained by a balance of constitutive vacuolar fusion and fission processes (*Baars et al., 2007*; *Li and Kane, 2009*).

The homotypic vacuolar membrane fusion process occurs in four stages: priming, tethering, docking, and fusion (*Wickner and Haas, 2000*; *Ostrowicz et al., 2008*; *Wickner, 2010*; *Qiu, 2012*). Each stage is defined as follows. 'Priming' In the priming reaction, inactive fusion factors that are still assembled from previous fusion events are recycled. The yeast *N*-ethylmaleimide-sensitive factor (NSF), Sec18p, is activated by ATP and releases the NSF attachment protein (SNAP), Sec17p, from the SNAP receptor (SNARE) complex, thereby activating the SNARE molecule (*Wickner and Haas, 2000*; *Ostrowicz et al., 2008*; *Müller et al., 2002*). 'Tethering' The HOPS/Class C Vps complex and the Rab-GTPase Ypt7p mediate the reversible contact of opposing vacuoles (*Ostrowicz et al., 2008*; *Ho and Stroupe, 2016*). 'Docking' The coupling of three Q-SNAREs (Vam3p, Vam7p, and Vti1p) with an opposing R-SNARE (Nyv1p) to form a trans-SNARE complex between the two corresponding vacuoles. 'Fusion' The final stage in which the lipid bilayer fuses and the contents of the lumen are mixed (*Wickner, 2010*; *Müller et al., 2002*). For stable vacuolar fusion, several factors are required such as vacuolar acidification by V-ATPase proton pump activity (*Baars et al., 2007*; *Ungermann et al., 1999*; *Peters et al., 2001*; *Desfougères et al., 2016*), ions (*Starai et al., 2005*), and ion transporters (*Qiu and Fratti, 2010*).

Furthermore, vacuolar fission is required for proper vacuolar inheritance during mitosis and acute response to osmotic shock in yeast, however the molecular mechanism of fission is not fully understood, and limited knowledge is available. Vacuolar fission is performed in two steps that require proteins and lipids (*Zieger and Mayer, 2012*). The first step is the contraction and invagination of the vacuolar membrane through the involvement of Vps1p, a dynamin-like GTPase (*Peters et al., 2004*) and V-ATPase, which drives the proton gradient (*Baars et al., 2007*). Next, vacuolar fission is promoted by Vac14p, Vac7p, and Fab1p (*Weisman, 2006*), which are required for the generation of PtdIns[3,5]P2, and Atg18p, an effector of Fab1p and a sensor of PtdIns[3,5]P2 levels (*Efe et al., 2005*; *Efe et al., 2007*). Recently, it has also been shown that vacuolar fission is associated with nutrient status and responds to endoplasmic reticulum (ER) stress via target of rapamycin complex 1 (TORC1) targets (*Michaillat et al., 2012*; *Stauffer and Powers, 2015*). Moreover, it has been proposed that vacuolar fusion and fission dynamics are regulated by the Yck3p-Env7p kinase cascade, which maintains an equilibrium between fusion and fission activities (*Manandhar et al., 2020*).

Organelles construct a dynamic network by associating with each other through membrane contact sites (MCSs), which play an important role in lipid transport and metabolism (*Wu et al., 2018*; *Prinz et al., 2020*). Lipids (phosphatidic acid (PA), diacylglycerol, ergosterol, and phosphatidylinositol (PtdIns) phosphate (PtdInsP)) and lipid raft domains have been shown to be important as fusion and fission effectors and discussed in a variety of recent reviews (*Li and Kane, 2009*; *Efe et al., 2005*; *Wickner and Rizo, 2017*; *Starr and Fratti, 2019*; *Ungermann and Kümmel, 2019*; *Hurst and Fratti, 2020*). However, it remains unclear if MCS-mediated lipid metabolism controls the morphology of the vacuole. Vacuoles have contact sites with other organelles. The MCS where the nuclear ER meets the vacuole is called the nucleus–vacuole junction (NVJ), and the ER-localized Nvj1p and vacuole-membrane-localized Vac8p act as tethers to form the NVJ (*Pan et al., 2000*). Vacuoles also form MCS with mitochondria called vacuole and mitochondria patches (vCLAMP). vCLAMP is mediated by the soluble (cytosol-localized) HOPS complex subunit Vps39p, the vacuole-localized small Rab-GTPase Ypt7p, and by an unknown mitochondrial factor (*Elbaz-Alon et al., 2014*; *Hönscher et al., 2014*). The translocase of the mitochondrial outer membrane (TOM) subunit Tom40p has been identified as a direct binding partner of Vps39p, suggesting a mechanism for the formation of vCLAMP by the Ypt7p-Vps39p-Tom40p tether (*González Montoro et al., 2018*). Although it has been suggested that MCSs with vacuoles are involved in the transport of lipids such as sterols and precursors of sphingolipids that are involved in vacuolar morphology (*Murley et al., 2015*; *Murley et al., 2017*; *Reinisch and Prinz, 2021*; *Girik et al., 2022*), the roles of those MCS-mediated lipid metabolisms in the regulation of vacuole morphology are not understood.

In this study, we found that tether proteins of MCS formation are involved in the regulation of vacuolar morphology. We show that the deletion of tricalbins, tethers of the MCS between the ER and the plasma membrane (PM) (*Manford et al., 2012*) and between the ER and the Golgi apparatus (*Ikeda et al., 2020*), affects vacuolar division. Phytosphingosine (PHS), a precursor of ceramide, accumulated in the tricalbin-deleted strain and was revealed as the underlying cause that triggers vacuolar fission. Moreover, deletion of key tethering proteins at the NVJ recovered vacuolar morphology of cells subjected to high exogenous PHS and tricalbin-deleted cells, indicating that the NVJ is required

for PHS-induced vacuolar fission. In summary, we propose that vacuolar morphology is regulated by MCSs through regulation of sphingolipid metabolic pathways.

## Results

### Deletion of tricalbins causes vacuole fragmentation

Tricalbins (Tcb1p, Tcb2p, and Tcb3p) are ER membrane tethering proteins that connect the cortical ER with the PM (*Toulmay and Prinz, 2012*), and contribute to PI4P turnover (*Manford et al., 2012*), sterol flux and transport (*Quon et al., 2018*) and maintain other lipid homeostasis (*Jorgensen et al., 2020*). It is also suggested that tricalbins create ER membrane curvature to maintain PM integrity (*Hoffmann et al., 2019*; *Collado et al., 2019*). Tricalbins also localize to the ER–Golgi contacts and are responsible for the non-vesicular transport of ceramide from ER to Golgi apparatus (*Ikeda et al., 2020*). However, their role in other organelle morphology and function remains unknown. Interestingly, the *NVJ1* gene, which encodes a major component of the NVJ, showed a negative synthetic interaction with the absence of all three tricalbins (*tcb1Δ2Δ3Δ*) (*Hoffmann et al., 2019*). Moreover, HOPS subunit Vam6p and Rab-GTPase Ypt7p, which are both involved in vacuolar fusion, also showed negative synthetic interaction with *tcb1Δ2Δ3Δ* (*Hoffmann et al., 2019*). Based on the findings, we assumed that tricalbins may be involved in the regulation of vacuolar morphology or function linked with the NVJ. To investigate the role of tricalbins in vacuole morphology, we analyzed the number of vacuoles per cell by a fluorescent probe FM4-64 that selectively stains yeast vacuolar membranes. We observed that compared to wild type cells, the *tcb1Δ2Δ3Δ* strain showed a phenotype characterized by a decreased percentage of cells with one vacuole and an increased percentage of cells with two or more vacuoles (*Figure 1A*). We refer to this phenotype as vacuolar fragmentation. In addition, analysis with single and double deletion strains revealed that single deletion of *TCB1* or *TCB3* already exhibited strong vacuole fragmentation (*Figure 1B*). These results indicate that tricalbins are important to maintain vacuole morphology.

We next examined whether deletion of tricalbins affects vacuolar acidification. As homotypic vacuole–vacuole membrane fusion requires the vacuolar H$^+$-ATPase (V-ATPase) (*Desfougères et al., 2016*; *Nelson and Nelson, 1990*; *Coonrod et al., 2013*), tricalbins could be indirectly involved in maintaining vacuole morphology through V-ATPase. Therefore, we addressed whether tricalbin mutant strains could grow in alkaline media, because V-ATPase-deficient mutants can grow in acidic media but fail to grow in alkaline media (*Nelson and Nelson, 1990*). As a negative control, we used *vma3Δ* mutant cells in which one of the V$_O$ subunits of V-ATPase had been deleted. The *vma3Δ* mutant cells grew in acidic medium (pH 5.0), but failed to grow in alkaline medium (pH 7.5) (*Figure 1— figure supplement 1A*). In contrast, all tricalbin mutant strains grew like wild type in both conditions, suggesting that the V-ATPase in these mutants remains functionally normal. Thus, these results suggest that vacuole fragmentation caused by tricalbin deletion is not due to a loss of function of V-ATPase.

Furthermore, we investigated the effects of tricalbin deletion on delivery of a vacuolar protease, carboxypeptidase S (Cps1p) that is sorted into the vacuole lumen upon endosome–vacuole fusion. Deletion of HOPS complex components that mediate endosome–vacuole fusion results in vacuole fragmentation (*Seeley et al., 2002*). To test the possibility that vacuole fragmentation in tricalbin mutant cells is caused by a defect of endosome–vacuole fusion, we analyzed the localization of GFP-Cps1p in tricalbin mutant cells. Cps1p, a vacuole-localized hydrolytic enzyme, is transported to the vacuole after sorting into multivesicular bodies by ESCRT (endosomal sorting complex required for transport) recognition (*Katzmann et al., 2001*). In strains with loss of ESCRT function, such as disruption of *VPS4*, one of the class E *VPS* (vacuolar protein sorting) genes, GFP-Cps1p is known to localize to the limiting membrane rather than the lumen of the vacuole (*Babst et al., 1997*; *Stuchell-Brereton et al., 2007*). In both WT and *tcb3Δ* cells, GFP-Cps1p was observed in the vacuole lumen in contrast to *vps4Δ* cells (*Figure 1—figure supplement 1B*), suggesting that the tricalbin mutant exhibits a normal delivery of vacuolar proteins via endosomes to the vacuole. In addition, the fact that tricalbin deletion does not affect the maturation of CPY, a vacuolar hydrolase carboxypeptidase Y (*Ikeda et al., 2020*), indicates that the vacuolar degradation ability as well as protein delivery to the vacuole is normal. Taken together, tricalbins are required for maintaining vacuole morphology, but not due to an indirect action through vacuolar acidification or endosome–vacuole fusion.

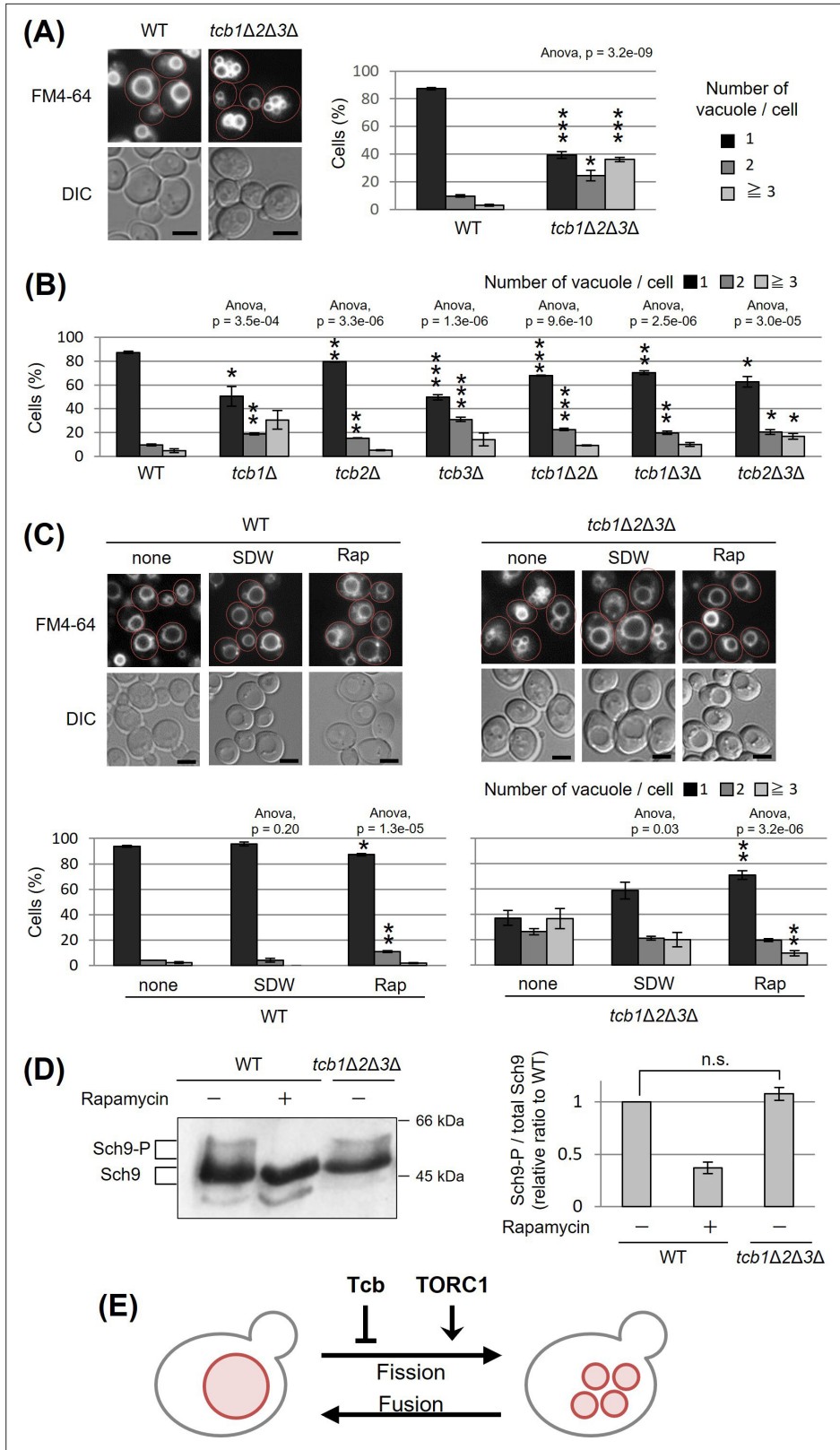

**Figure 1.** Deletion of tricalbin proteins causes vacuole fragmentation. (**A, B**) Cells (FKY2577 and FKY2927 in A; FKY2577, FKY2909, FKY3819, FKY2924, FKY3023, FKY3820, and FKY3008 in B) were grown overnight at 25°C in YPD. Then vacuoles were stained with FM4-64 and imaged by fluorescence microscopy. Scale bar, 5 mm. The number of vacuoles per cell was counted and categorized into one of three groups. The data represent mean

*Figure 1 continued on next page*

*Figure 1 continued*

± standard error (SE) of three independent experiments, each based on more than 100 cells. *p < 0.05, **p < 0.01, and ***p < 0.001 by Student's *t*-test compared with wild-type (WT). (**C**) Cells (FKY2577 and FKY2927) were grown overnight at 25°C in YPD. Cells were then incubated in sterile distilled water (SDW) for more than 45 min or YPD with 200 nM of rapamycin (Rap) for 2 hr. Vacuoles were stained with FM4-64 and imaged by fluorescence microscopy. Scale bar, 5 mm. The number of vacuoles per cell was counted and categorized into one of three groups. The data represent mean ± SE of more than three independent experiments, each based on more than 90 cells. *p < 0.05, **p < 0.01 by Student's *t*-test compared with none treated cells. (**A–C**) Significant differences analysis between the pairwise combination of groups was performed using two-way analysis of variance (ANOVA). (**D**) Cells (FKY2577 and FKY2927) transformed with pRS416-SCH9-5HA were cultured in YPD, treated with 200 nM rapamycin (control) or untreated. The extracts from cells expressing Sch9-5HA were reacted with 2-nitro-5-thiocyanobenzoic acid and analyzed by immunoblotting using anti-HA. Phosphorylated Sch9 relative to the total Sch9 was calculated and shown in comparison to untreated WT cells. The data represent mean ± SE of three independent experiments. n.s., not significant by Student's *t*-test. (**E**) Illustration shows that tricalbin proteins negatively regulate the vacuole fission in a target of rapamycin complex 1 (TORC1)-independent manner.

The online version of this article includes the following source data and figure supplement(s) for figure 1:

**Source data 1.** Excel file of numerical data represented as a graph in *Figure 1A*.

**Source data 2.** Excel file of numerical data represented as a graph in *Figure 1B*.

**Source data 3.** Excel file of numerical data represented as two graphs in *Figure 1C*.

**Source data 4.** Excel file of numerical data represented as a graph in *Figure 1D*.

**Source data 5.** Original file for the Western blot analysis in *Figure 1D*.

**Source data 6.** JPEG containing *Figure 1D* and original scans of the relevant Western blot analysis with highlighted bands and sample labels.

**Figure supplement 1.** Vacuolar acidification and Cps1p delivery are not affected by tricalbin deletion.

## Tricalbins negatively regulate vacuole fission in a parallel pathway with TORC1

The number and size of vacuoles within a cell are regulated by coordinated cycles of vacuolar fission and fusion. To address whether vacuole fragmentation in the tricalbin mutant is caused by facilitation of vacuole fission or an impaired homotypic vacuole fusion, we examined the effect of hypotonic stress on vacuole fragmentation in *tcb1Δ2Δ3Δ* mutant cells, because low-osmotic stimuli such as water promote homotypic vacuole fusion to maintain cytoplasmic osmolarity (*Dove et al., 2009*). We observed that addition of water to *tcb1Δ2Δ3Δ* cells partially restored vacuole fragmentation, suggesting that vacuolar fusion machinery is functional in *tcb1Δ2Δ3Δ* cells (*Figure 1C*). This result suggests that loss of tricalbins alters fusion–fission dynamics by primarily affecting the fission machinery rather than fusion.

As the TORC1 is a positive regulator of vacuole fission in response to hyperosmotic shock and ER stress (*Michaillat et al., 2012*; *Stauffer and Powers, 2015*), we next examined if rapamycin, which is an inhibitor of TORC1, affects the vacuole fragmentation in *tcb1Δ2Δ3Δ* cells. As shown in *Figure 1C*, we observed that the vacuole fragmentation in *tcb1Δ2Δ3Δ* cells was suppressed by rapamycin, suggesting that TORC1 may be required for tricalbin deletion-induced vacuolar fragmentation. We attempted to characterize further the relationship between tricalbin and TORC1, and showed that *tcb1Δ2Δ3Δ* cells had no significant effect on phosphorylation levels of Sch9p, a major downstream effector of TORC1 (*Figure 1D*). These results suggest that deletion of tricalbins does not activate TORC1. Therefore, we conclude that tricalbins and TORC1 act in parallel and opposite ways to regulate vacuole fission (*Figure 1E*).

## The transmembrane domain of Tcb3 contributes to mediating protein interactions between the tricalbin family to maintain vacuolar morphology

Tcb3p possesses an N-terminal transmembrane (TM) domain, a central synaptotagmin-like mitochondrial lipid-binding protein (SMP) domain, and multiple C-terminal $Ca^{2+}$-dependent lipid-binding (C2) domains (*Figure 2A*). To identify which domain of Tcb3p is essential for regulating vacuole fission, we replaced the C-terminal sequence of the endogenous *TCB3* gene with a GFP-binding protein

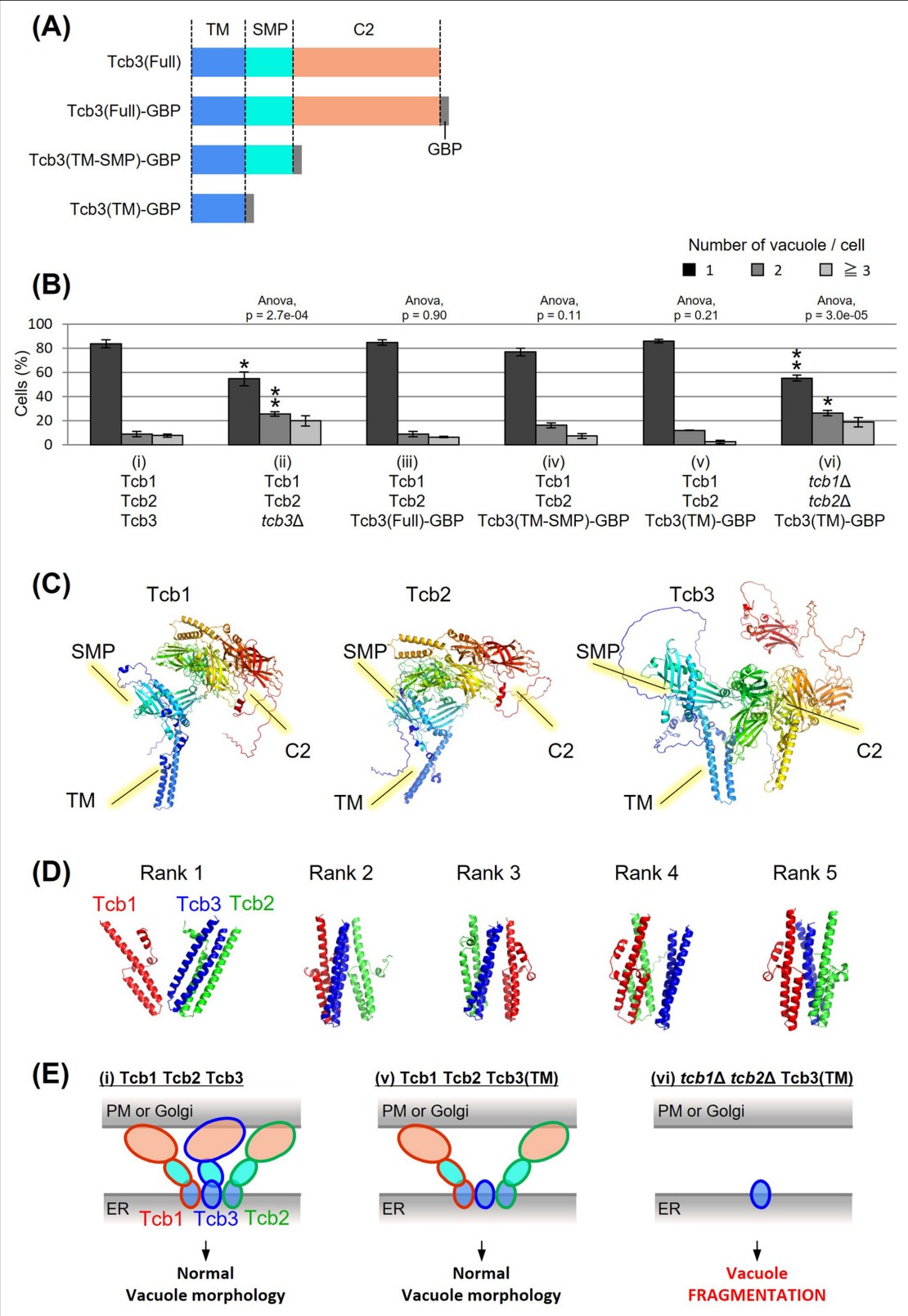

**Figure 2.** Effects of domain deletion on vacuole morphology. (**A**) Diagram of domain organization of Tcb3 protein. TM, transmembrane domain; SMP, synaptotagmin-like mitochondrial lipid-binding protein; C2, calcium-dependent lipid-binding domain; GBP, GFP-binding protein. (**B**) Cells (FKY2577 (i), FKY2924 (ii), FKY3903 (iii), FKY3904 (iv), FKY3905 (v), and FKY4754 (vi)) were grown overnight at 25°C in YPD. Then vacuoles were stained with FM4-64 and imaged by fluorescence microscopy. The number of vacuoles per cell was counted and categorized into one of three groups. The data represent

*Figure 2 continued on next page*

*Figure 2 continued*

mean ± standard error (SE) of three independent experiments, each based on more than 90 cells. *p < 0.05 and **p < 0.01 by Student's *t*-test compared with Tcb1 Tcb2 Tcb3 (i). Significant differences analysis between the pairwise combination of groups was performed using two-way analysis of variance (ANOVA). (**C**) The modeled structures of the Tcb1p, Tcb2p, and Tcb3p proteins. The ribbons and arrows in the models indicate alpha-helices and beta-sheets, respectively. (**D**) TM domain complex in Tcb1p (red), Tcb2p (green), and Tcb3p (blue). The rank 'X' indicates the order in which the complexes are most likely to form. (**E**) Illustrations show that TM domain of Tcb3 contributes to mediating protein interactions between the tricalbin family to maintain vacuolar morphology.

The online version of this article includes the following source data and figure supplement(s) for figure 2:

**Source data 1.** Excel file of numerical data represented as a graph in *Figure 2B*.

**Figure supplement 1.** Coimmunoprecipitation assay between Tcb3-HA and Tcb1-GFP or Tcb2-GFP.

**Figure supplement 1—source data 1.** Original file for the Western blot analysis in *Figure 2—figure supplement 1* (anti-HA).

**Figure supplement 1—source data 2.** Original file for the Western blot analysis in *Figure 2—figure supplement 1* (anti-GFP).

**Figure supplement 1—source data 3.** JPEG containing *Figure 2—figure supplement 1* and original scans of the relevant Western blot analysis (anti-HA and anti-GFP) with highlighted bands and sample labels.

(GBP). This way, we created three strains expressing either full-length Tcb3-GBP; Tcb3(Full)-GBP, Tcb3-GBP lacking C2 domains; Tcb3(TM-SMP)-GBP or Tcb3-GBP lacking both SMP and C2 domains; Tcb3(TM)-GBP. Vacuole morphology in cells expressing Tcb3(Full)-GBP (*Figure 2B* (iii)) was almost similar to that in WT cells (*Figure 2B* (i)), indicating that the addition of GBP has no effect on vacuole morphology. Remarkably, vacuoles remained non-fragmented even in cells expressing Tcb3(TM)-GBP, which lacked most of the C-terminal region (*Figure 2B* (v)), as well as Tcb3(TM-SMP)-GBP (*Figure 2B* (iv)). In an attempt to address the question of why the TM domain of Tcb3p is sufficient to suppress vacuolar division, we found that cells expressing Tcb3(TM)-GBP and lacking Tcb1p and Tcb2p (*Figure 2B* (vi)) are even more fragmented than *tcb1Δ2Δ* in *Figure 1B*, and similar to *tcb3Δ* (*Figure 1B* and *Figure 2B* (ii)). These results suggest that the TM domain of Tcb3p requires Tcb1p and Tcb2p to suppress vacuole fission. Tcb2p has been reported to interact with either Tcb1p or Tcb3p through their C-terminal domain (*Creutz et al., 2004*), and some of the protein interactome analyses have reported that Tcb1p, Tcb2p, and Tcb3p interact with each other (*Tarassov et al., 2008*; *Michaelis et al., 2023*). In this study, we have also confirmed that Tcb3 shows physical interaction with both Tcb1 and Tcb2 by the coimmunoprecipitation assay (*Figure 2—figure supplement 1*). In addition, according to the structural simulation by AlphaFold2, each TM domain of Tcb1p, Tcb2p, and Tcb3p formed alpha-helical structure, which is known to be typical for membrane proteins (*Figure 2C*). It is likely that Tcb1p and Tcb2p directly interact with Tcb3p by their TM domains. Interestingly, the complex of these three TM domains was constructed with Tcb3p between Tcb1p and Tcb2p in most cases (*Figure 2D*). Together these data suggest that the TM domain of Tcb3p contributes sufficiently to the maintenance of vacuolar morphology by mediating the tricalbin complex formation (*Figure 2E*).

## Accumulated PHS in tricalbin-deleted cells causes vacuole fragmentation

To understand how the deletion of tricalbins leads to vacuole fragmentation, we examined the involvement of lipids. Previous reports showed that levels of acylceramides, which are made by adding another acyl chain to ceramides, increased in *tcb1Δ2Δ3Δ* cells (*Ikeda et al., 2020*), and levels of long-chain bases (LCBs) increased in Δtether cells lacking six ER–PM tethering proteins (*Omnus et al., 2016*). Here, we measured lipids in *tcb1Δ2Δ3Δ* cells by in vivo labeling with [$^3$H] dihydrosphingosine (DHS), which is a precursor of PHS, and observed significant increases in ceramide species, phosphatidylethanolamine, PHS, phosphatidylinositol, complex sphingolipids such as inositolphosphorylceramide (IPC) and mannosyl-inositolphosphorylceramide (MIPC) and LCB-1P (DHS-1P/PHS-1P) levels (*Figure 3A*).

We first evaluated whether the increases in certain lipids are the cause of vacuolar fragmentation in *tcb1Δ2Δ3Δ*. Our analysis showed that vacuoles are fragmented in *lag1Δlac1Δ* cells, which lack both enzymes for LCBs (DHS and PHS) conversion into ceramides (*Figure 3B*). Loss of ceramide synthases could cause an increase in PHS levels. Therefore, we tested if exogenously added PHS induces vacuolar fragmentation in WT cells. As shown in *Figure 3C*, exogenous addition of PHS-induced vacuolar fragmentation. PHS is converted into ceramide by the ceramide synthase Lag1p, which is the main enzyme synthesizing phytoceramide (*Megyeri et al., 2019*). Thus, we next examined whether

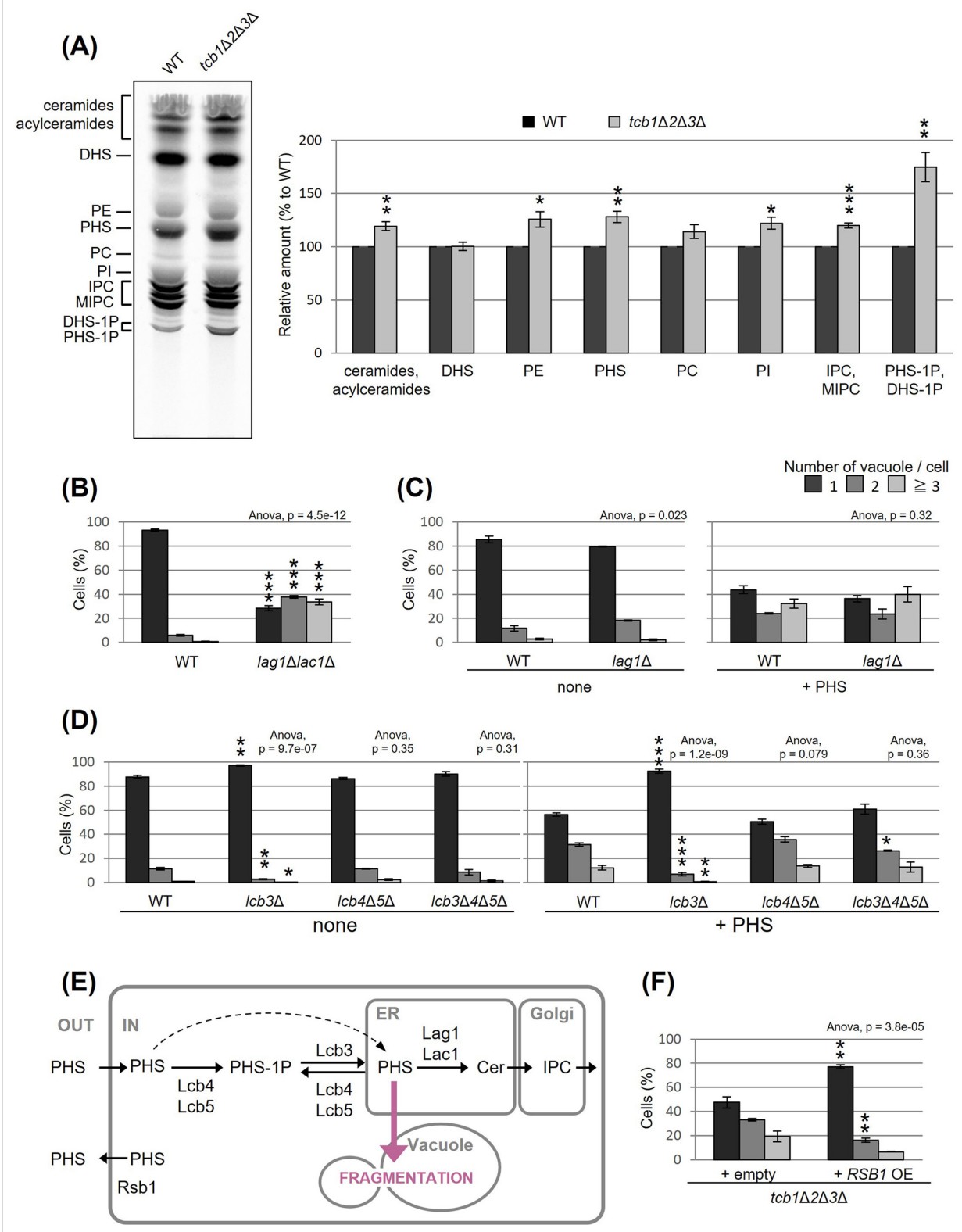

**Figure 3.** Accumulated phytosphingosine (PHS) in *tcb1Δ2Δ3Δ* causes vacuole fragmentation. (**A**) Cells (FKY2577 and FKY2927) were grown at 25°C, and labeled with [³H]DHS for 3 hr. Labeled lipids were applied to thin-layer chromatography (TLC) plates using solvent system (chloroform-methanol-4.2N ammonium hydroxide (9:7:2, vol/vol/vol)). Incorporation of [³H]DHS into each lipid was quantified and the percentage of the total radioactivity (%) in WT cells was determined. Data represent mean ± standard error (SE) of four independent experiments. **p < 0.01 by Student's *t*-test. (**B–D, F**) Cells (FKY5687 and FKY5688 in B; FKY3340 and YKC121-59 in C; FKY36, FKY37, FKY33, and FKY38 in D; FKY2927 in F) were grown overnight at 25°C in YPD.

*Figure 3 continued on next page*

*Figure 3 continued*

PHS was added at 160 µM (**C**) or 80 µM (**D**) for 2 hr. Vacuoles were stained with FM4-64 and imaged by fluorescence microscopy. The number of vacuoles per cell was counted and categorized into one of three groups. The data represent mean ± SE of three independent experiments, each based on more than 100 cells. *p < 0.05, **p < 0.01, and ***p < 0.001 by Student's *t*-test compared with WT (**B, D**) or empty cells (**F**). Significant differences analysis between the pairwise combination of groups was performed using two-way analysis of variance (ANOVA). (**E**) Illustration showing intracellular utilization pathway of exogenous PHS.

The online version of this article includes the following source data for figure 3:

**Source data 1.** Original file for the thin-layer chromatography (TLC) analysis in *Figure 3A*.

**Source data 2.** JPEG containing *Figure 3A* and original scans of the relevant thin-layer chromatography (TLC) analysis with highlighted bands and sample labels.

**Source data 3.** Excel file of numerical data represented as a graph in *Figure 3A*.

**Source data 4.** Excel file of numerical data represented as a graph in *Figure 3B*.

**Source data 5.** Excel file of numerical data represented as two graphs in *Figure 3C*.

**Source data 6.** Excel file of numerical data represented as two graphs in *Figure 3D*.

**Source data 7.** Excel file of numerical data represented as a graph in *Figure 3F*.

PHS-induced vacuolar fragmentation occurs in *lag1Δ* cells. Our analysis showed that vacuolar fragmentation in *lag1Δ* cells treated with PHS was comparable to that for WT cells (*Figure 3C*). These results suggest that the increases in ceramide and subsequent product IPC/MIPC are not the cause of vacuolar fragmentation, but rather its precursors induce vacuolar fragmentation.

Because our lipid analysis showed a strong increase in LCB-1P in *tcb1Δ2Δ3Δ* cells, phosphorylation of PHS may be involved in PHS induced vacuolar fragmentation. Although exogenously added LCBs move slowly from the PM to the ER if they are not phosphorylated, efficient utilization of exogenously added LCBs in sphingolipid synthesis requires a series of phosphorylation and dephosphorylation steps for LCBs (*Funato et al., 2003*). The major yeast LCB phosphate phosphatase Lcb3p dephosphorylates LCB-1P (DHS-1P/PHS-1P) to yield DHS/PHS (*Qie et al., 1997*), while Lcb4p and Lcb5p catalyze the reverse reaction producing DHS-1P/PHS-1P (*Nagiec et al., 1998*). Lcb3p, Lcb4p, and Lcb5p are localized to the ER, PM, or Golgi (*Funato et al., 2003*; *Mao et al., 1999*; *Hait et al., 2002*; *Iwaki et al., 2007*). We examined the ability of exogenous PHS to fragment vacuoles in cells lacking these processing factors. Our results showed that PHS-induced vacuolar fragmentation was completely blocked in the *lcb3Δ* cells in which PHS-1P is not dephosphorylated (*Figure 3D*), suggesting that PHS-induced vacuolar fragmentation requires the reaction of dephosphorylation of PHS-1P by Lcb3p. On the other hand, we observed that vacuoles were still fragmented in *lcb4Δ lcb5Δ* and *lcb3Δ lcb4Δ lcb5Δ* cells (*Figure 3D*). These results suggest that elevated levels of non-phosphorylated PHS induces vacuolar fragmentation (*Figure 3E*). Additionally, we tested whether overexpression of Rsb1p, which has been reported as a translocase that exports LCBs from the inner to the outer leaflet of the PM (*Kihara and Igarashi, 2002*), rescues vacuole fragmentation in *tcb1Δ2Δ3Δ* cells. As shown in *Figure 3F*, we observed that Rsb1p overexpression results in decreased vacuole fragmentation. Collectively, these results support the model in which vacuole fragmentation in tricalbin-deleted cells is caused by increased levels of PHS.

## The NVJ is required for PHS- or *tcb3Δ*-induced vacuole fragmentation

To characterize further the relationship between PHS and vacuole morphology, we asked whether PHS delivered from the ER to the vacuole induces vacuole fragmentation. We hypothesized that if PHS is transported to the vacuole by either the vesicular transport pathway through the Golgi apparatus or the non-vesicular transport pathway via inter-organellar MCS triggers vacuole fission, then blockage of the transport could rescue the PHS-induced vacuole fragmentation. Accordingly, we used two types of mutant strains, one blocking vesicular transport and the other blocking the NVJ. In a temperature-sensitive *sec18-20* mutant, vesicular transport is abolished at the restrictive temperature of 30°C or higher, but even at the permissive temperature of 25°C, vesicular transport is partially impaired (*Funato and Riezman, 2001*). As shown in *Figure 4A* (left), under the condition without exogenous PHS, some vacuoles in *sec18-20* mutant cells became fragmented when shifted to the non-permissive temperature of 30°C. We also observed that vacuoles in *sec18-20* mutant at 30°C underwent fragmentation after addition of PHS to the same extent as at the permissive temperature

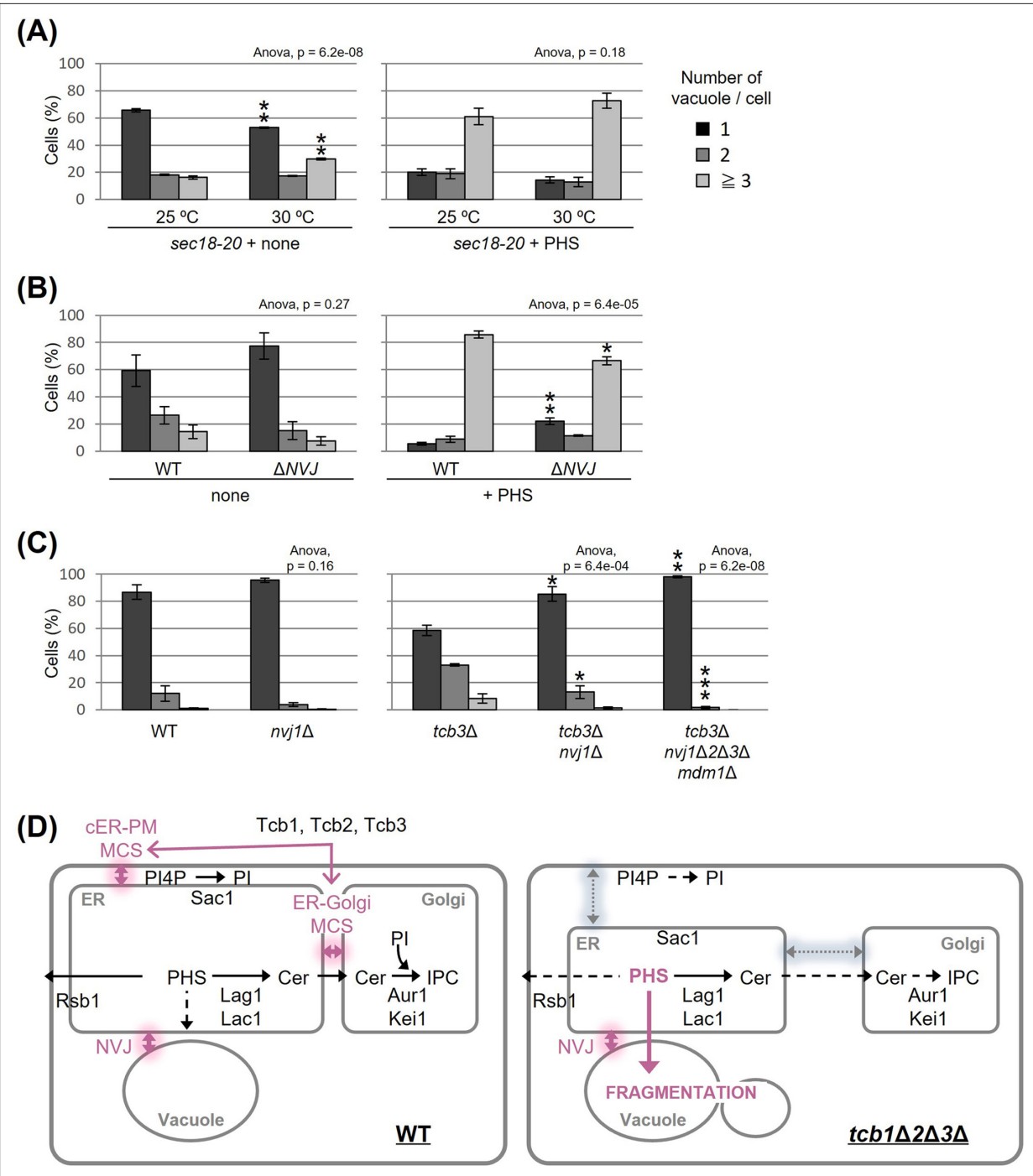

**Figure 4.** Nucleus–vacuole junction (NVJ) is required for phytosphingosine (PHS)-induced vacuole fragmentation. (**A–C**) Cells (FKY2929 in A; FKY3868 and FKY5560 in B; FKY6187, FKY6189, FKY6190, FKY6188, and FKY6409 in C) were grown overnight at 25°C in YPD. PHS was added at 40 µM for 2 hr at 30°C (**A**) and 25°C (**A, B**). Vacuoles were stained with FM4-64 and imaged by fluorescence microscopy. The number of vacuoles per cell was counted and categorized into one of three groups. The data represent mean ± standard error (SE) of three independent experiments, each based on more than 100 cells. *$p < 0.05$, **$p < 0.01$, and ***$p < 0.001$ by Student's $t$-test. Significant differences analysis between the pairwise combination of groups was performed using two-way analysis of variance (ANOVA). (**D**) Membrane contact sites regulate vacuole morphology via sphingolipid metabolism. See the main text for details.

The online version of this article includes the following source data and figure supplement(s) for figure 4:

**Source data 1.** Excel file of numerical data represented as two graphs in *Figure 4A*.

**Source data 2.** Excel file of numerical data represented as two graphs in *Figure 4B*.

*Figure 4 continued on next page*

*Figure 4 continued*

**Source data 3.** Excel file of numerical data represented as two graphs in *Figure 4C*.

**Figure supplement 1.** Cells (FKY3340, YKC145-21, and YKC149-61) were grown overnight at 25°C in YPD.

**Figure supplement 1—source data 1.** Excel file of numerical data represented as two graphs in *Figure 4—figure supplement 1*.

of 25°C (*Figure 4A*, right), suggesting that vesicle-mediated transport is not required for PHS-induced vacuolar fragmentation. To test the role of NVJ-mediated membrane contact for PHS-induced vacuolar fragmentation we employed a quadruple Δ*NVJ* mutant (*nvj1Δ nvj2Δ nvj3Δ mdm1Δ*) that was used in a previous study in which complete loss of the NVJ was observed (*Henne et al., 2015*), but thus has a different background to other strains of this study. When PHS was added to the Δ*NVJ* mutant, we observed a significant suppression of vacuole fragmentation compared to WT (*Figure 4B*). Finally, to investigate the requirement for NVJ in tricalbin deletion-induced vacuolar fragmentation, we constructed the *tcb3Δ nvj1Δ* and *tcb3Δ nvj1Δ nvj2Δ nvj3Δ mdm1Δ* mutants. *TCB3* single disruption sufficiently induced vacuolar fragmentation (*Figure 1B*), whereas as expected, the fragmentation was partially suppressed by loss of only *NVJ1* and completely suppressed by loss of all NVJ factors (*NVJ1*, *NVJ2*, *NVJ3*, and *MDM1*) (*Figure 4C*). Taken together, we conclude from these findings that vacuole fission caused by accumulated PHS in tricalbin-deleted cells requires contact between ER and vacuole at the NVJ, possibly as a way to transport PHS to the vacuole.

## NVJ and PHS accumulation mediate hyperosmotic shock-induced vacuole fission

Besides PHS-induced vacuolar fission, it is generally well known that vacuolar division can be triggered as an acute response to osmotic shock. We made an interesting observation under hyperosmotic conditions (0.2 M NaCl), wherein the loss of NVJ led to complete suppression of vacuolar division (*Figure 5A*). This finding suggests a significant role for NVJ in vacuolar fission as a hyperosmotic response. To test the involvement of PHS accumulation in this process, we analyzed the effect of hyperosmolarity on PHS levels. Analysis of PHS under hyperosmotic shock conditions (0.2 M NaCl), in which vacuolar fragments were observed, showed an increase in PHS of about 10% (*Figure 5B*). Furthermore, when the NaCl concentration was increased to 0.8 M, PHS levels increased up to 30%. While NaCl treatment increased PHS, both ceramide and IPC decreased. There are at least two possible reasons why NaCl increases PHS and decreases ceramide and IPC. The first is the possibility that NaCl dissociates subunits of ceramide synthase (Lag1p, Lac1p, and Lip1p) or IPC synthase (Aur1p and Kei1p). The second possibility is that NaCl suppresses the expression of ceramide synthase or IPC synthase, as can be inferred from the previous report (*Manzanares-Estreder et al., 2017*). Alternatively, we cannot exclude the possibility that the difference in PHS levels detected with [³H]DHS in this study is due to differences in the activity of Sur2p hydroxylase that catalysis the conversion of DHS to PHS (*Haak et al., 1997*). Finally, NaCl-induced vacuolar fragmentation, like that caused by PHS treatment, was also suppressed by PHS export from the cell by Rsb1p overexpression (*Figure 5C*). These results suggest that hyperosmotic shock-induced vacuole fission is also mediated by PHS accumulation and NVJ.

## Discussion

In the present study, we found that the accumulation of PHS triggers the fission of vacuoles. Our results suggest that MCSs are involved in this process in two steps. First, the intracellular amount of PHS is modulated by tricalbin-tethered MCSs between the ER and PM or Golgi (*Figure 4D*, left). Second, the accumulated PHS in the tricalbin-deleted cells induces vacuole fission via most likely the NVJ (*Figure 4D*, right). Thus, we propose that MCSs regulate vacuole morphology via sphingolipid metabolism. Fundamental questions that arise from our data concern the accumulation of PHS in the tricalbin deletion strain and the mechanism of PHS-induced vacuole fission. Possible mechanisms for elevated PHS levels in tricalbin-deleted cells are the following. MCS deficiency between ER and PM has been shown to reduce the activity of Sac1p, a PtdIns4P phosphatase (*Manford et al., 2012*). Sac1p disruption strain decreases the levels of complex sphingolipids such as IPC and MIPC, while it increases the levels of their precursors, ceramide, LCB, and LCB-1P (*Brice et al., 2009*). This is

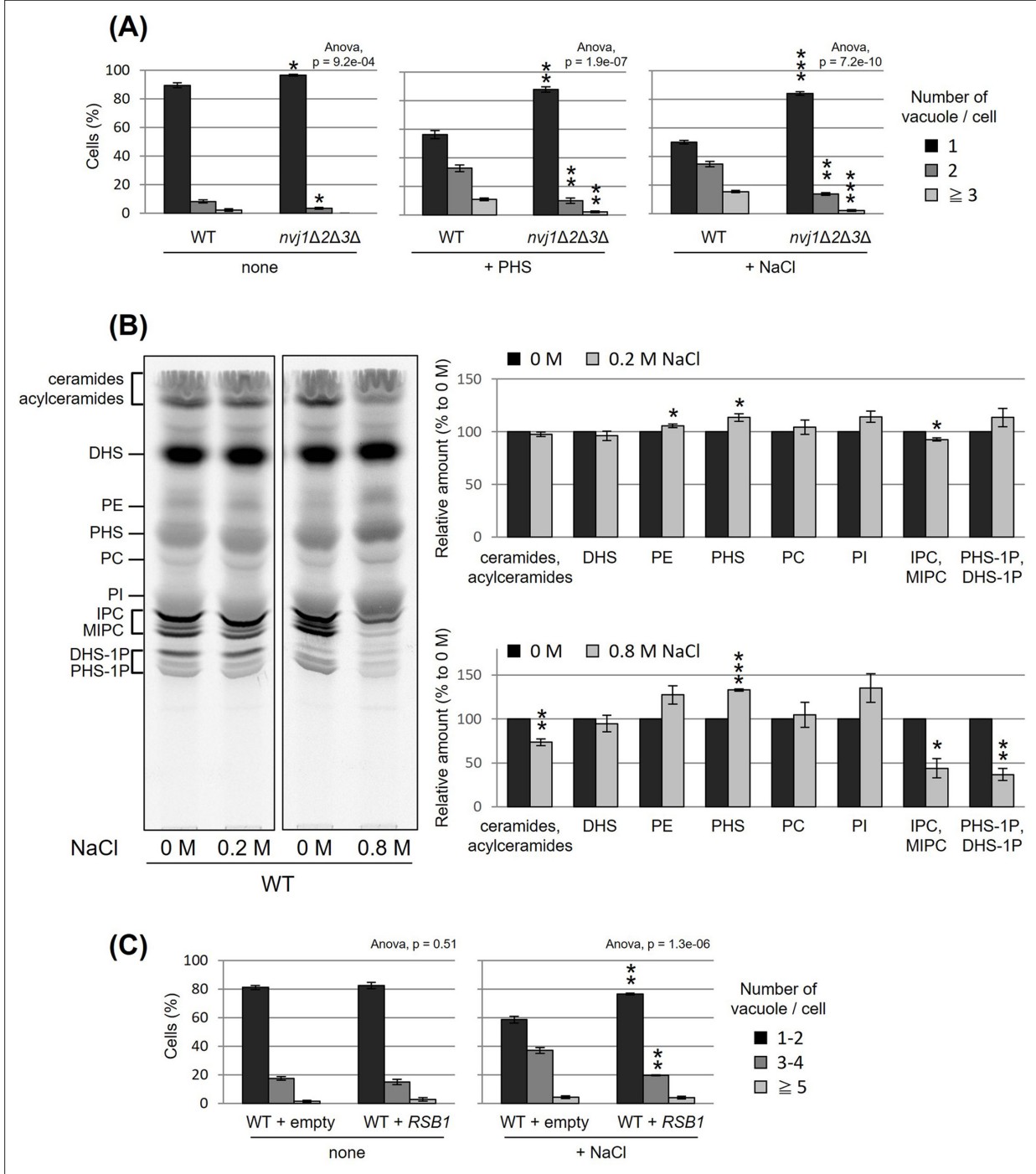

**Figure 5.** Nucleus–vacuole junction (NVJ) and phytosphingosine (PHS) accumulation mediate hyperosmotic shock-induced vacuole fission. (**A**) Cells (FKY6187 and FKY6140) were grown overnight at 25°C in YPD, incubated with 80 μM of PHS or 0.2 M of NaCl for 2 hr. Vacuoles were stained with FM4-64 and imaged by fluorescence microscopy. The number of vacuoles per cell was counted and categorized into one of three groups. (**B**) Cells (FKY2577) were grown at 25°C then labeled with [³H]DHS and incubated with 0.2 or 0.8 M of NaCl for 2 hr. Labeled lipids were applied to thin-layer chromatography (TLC) plates using solvent system (chloroform-methanol-4.2N ammonium hydroxide (9:7:2, vol/vol/vol)). Incorporation of [³H]DHS into each lipid was quantified and the percentage of the total radioactivity (%) in WT cells was determined. Data represent mean ± standard error (SE) of four independent experiments. (**C**) Cells (FKY2577) were grown overnight at 25°C in SD, then incubated with 0.2 M of NaCl for 2 hr. Vacuoles were stained with FM4-64 and imaged by fluorescence microscopy. The number of vacuoles per cell was counted and categorized into one of three groups. (**A–C**) The data represent mean ± SE of three independent experiments. *p < 0.05, **p < 0.01, and ***p < 0.001 by Student's t-test. Significant differences analysis between the pairwise combination of groups was performed using two-way analysis of variance (ANOVA).

*Figure 5 continued on next page*

*Figure 5 continued*

The online version of this article includes the following source data for figure 5:

**Source data 1.** Excel file of numerical data represented as three graphs in *Figure 5A*.

**Source data 2.** Original file for the thin-layer chromatography (TLC) analysis in *Figure 5B*.

**Source data 3.** JPEG containing *Figure 5B* and original scans of the relevant thin-layer chromatography (TLC) analysis with highlighted bands and sample labels.

**Source data 4.** Excel file of numerical data represented as two graphs in *Figure 5B*.

**Source data 5.** Excel file of numerical data represented as two graphs in *Figure 5C*.

probably due to the loss of function of Sac1p, which reduces the level of PtdIns used for IPC synthesis, thereby impairing IPC synthesis, and resulting in the accumulation of precursors such as the substrate ceramide. This model is also consistent with the results of accumulated PtdIns4P and reduced IPC synthesis in Δtether cells (*Omnus et al., 2016*). As tricalbins are required for ceramide non-vesicular transport from the ER to the Golgi (*Ikeda et al., 2020*), it is also possible that impaired ceramide non-vesicular transport due to tricalbin deficiency causes ceramide and its precursor, LCB, to accumulate in the ER. Another possibility is that MCS facilitate PHS diffusion between the ER and the PM, which might be coordinated with the LCB export from the PM by Rsb1p. The loss of tricalbins partially disrupts the ER–PM tether, possibly resulting in low efficiency of PHS ejection by Rsb1p. This is supported by the result that overexpression of Rsb1p suppressed vacuolar fragmentation in tricalbin-deleted cells (*Figure 3F*).

Previous studies have observed that myriocin treatment results in vacuolar fragmentation (*Fröhlich et al., 2015*; *Hepowit et al., 2023*). Myriocin treatment itself causes not only the depletion of PHS but of complex sphingolipids such as IPC. This suggests that normal sphingolipid metabolism is important for vacuolar morphology. The reason for this is unclear, but perhaps there is some mechanism by which sphingolipid depletion affects, for example, the recruitment of proteins required for vacuolar membrane fusion. In contrast, our new findings show that PHS increase causes vacuole fragmentation. Taken together, there may be multiple mechanisms controlling vacuole morphology by both increasing and decreasing PHS.

Based on the fact that both PHS- and tricalbin deletion-induced vacuolar fragmentations were partially suppressed by the lack of NVJ (*Figure 4B, C*), it is possible that the trigger for vacuolar fragmentation is NVJ-mediated transport of PHS into the vacuole. Recently, it has been reported that sphingoid bases are transferred between ER and vacuole via the NVJ, and that Mdm1p, a key tethering protein for the formation of the NVJ, may play an additional role in LCB transfer (*Girik et al., 2022*). The study applied an experimental method that tracks LCBs released in the vacuole and showed that Mdm1p is necessary for LCBs leakage into the ER. However, assuming that Mdm1p transports LCBs along its concentration gradient we consider that under normal conditions, LCBs is transported from the ER (as the organelle of PHS synthesis) to the vacuole. Perhaps, Mdm1p may be responsible for pulling out and passing LCBs. However, we cannot rule out the possibility that the repression of vacuolar fragmentation in the absence of NVJ is not due to inhibition of PHS transfer, but rather to changes in the lipid composition of the vacuolar membrane caused by the lack of supply of other substances capable of triggering vacuolar fragmentation other than PHS, like sterols and lipids such as PtdIns[3,5]P2 and its precursors. Further analysis in this regard is warranted.

How accumulated PHS triggers vacuolar fragmentation remains undetermined. Fab1p, a target of TORC1, is responsible for the production of PtdIns[3,5]P2, which is a well-established inducer of vacuolar fragmentation. Fab1p exhibits co-localization with the TORC1-activating EGO complex, and its activity is controlled by Ivy1p (*Malia et al., 2018*) and TORC1 (*Chen et al., 2021*). PtdIns[3,5]P2 was shown to regulate vacuole fission employing Vps1p and Atg18p as executioners (*Gopaldass et al., 2017*) while also promoting TORC1 activity in a positive feedback loop (*Jin et al., 2014*). In this context, we found that PHS-induced vacuolar fragmentation can be suppressed by the loss of Fab1p and its regulatory binding partner Vac14p (*Figure 4—figure supplement 1*). This observation suggests that PHS-induced vacuolar fragmentation would employ known factors such as Fab1p or Vac14p as the executioners.

The following possibilities still remain, although less likely than the above. Sphingosine, as a bioactive lipid, has been reported to exert effects on enzyme activity in humans and yeast (*Hannun*

*et al., 1986*; *Chang et al., 2001*; *Yabuki et al., 2019*), and it is possible that PHS may induce vacuolar fragmentation through established signaling pathways. *Yabuki et al., 2019* reported that LCB accumulation activates a signaling pathway that includes major yeast regulatory kinases such as Pkh1/2p, Pkc1p, and TORC1, which may be a candidate to promote vacuolar fragmentation. On the other hand, it was shown that membrane division is mediated by certain proteins that contain amphipathic helices (AHs) and interact with lipid cofactors such as PtdIns[4,5]P2, PA, and cardiolipin (*Zhukovsky et al., 2019*). Thus, PHS may possess a similar regulatory function as a lipid cofactor for the activity of fission-inducing proteins. If PHS-induced vacuole fragmentation is not due to signaling, another possible model is that PHS accumulation in the vacuolar membrane causes physical changes in the membrane structure that result in membrane fragmentation. The accumulation of sphingosine in the GARP mutant *vps53Δ*, in which retrograde transport from the endosome to the Golgi is blocked, caused vacuolar fragmentation (*Fröhlich et al., 2015*), and thus also supports this model. Sphingosine stabilizes (rigidifies) the gel domains in the membrane, leading to a structural defect between the phase separation of 'more rigid' and 'less rigid' domains (*Contreras et al., 2006*). This structural defect may result in high membrane permeability. Sphingosine also forms small and unstable channels (compared to the channels formed by ceramide) in the membrane (*Siskind et al., 2005*). Sphingosine channels are not large enough to release proteins but are believed to induce a permeability transition. Other studies have suggested that sphingosine induces non-lamellar structures by interacting with negatively charged lipids such as PA (*Jiménez-Rojo et al., 2014*). PHS alone or in concert with negatively charged lipids such as PtdIns[3,5]P2 may be actively involved in the fission process by inducing structural changes in the vacuolar membrane.

Finally, additional results provided further insight into the more general aspects of PHS involvement in the vacuole fission process. Lipid analysis under hyperosmotic shock condition (0.2 or 0.8 M of NaCl) showed an increase in PHS level (*Figure 5B*). NaCl-induced vacuolar fragmentation was also suppressed by Rsb1p-mediated PHS export (*Figure 5C*), as was NVJ loss (*Figure 5A*). In addition to the hyperosmotic shock-induced PHS accumulation, we have previously shown that treatment with tunicamycin, which is ER stress inducer, increased the PHS level by about 20% (*Yabuki et al., 2019*). Tunicamycin treatment has been shown to induce vacuole fission (*Stauffer and Powers, 2015*). Collectively, we propose that the NVJ and PHS play a general regulatory role in vacuolar morphology.

## Materials and methods

### Yeast strains

All strains of *Saccharomyces cerevisiae* used for this work are listed in *Supplementary file 1*.

### Plasmids

All plasmids used for this work are listed in *Supplementary file 2*.

### Culture conditions

Yeast cells were grown either in rich YPD medium (2% glucose, 1% yeast extract, 2% peptone) or in synthetic minimal SD medium (2% glucose, 0.15% yeast nitrogen base, 0.5% ammonium sulfate, bases as nutritional requirements) and supplemented with the appropriate amino acids.

### FM4-64 stain and fluorescence microscopy

Yeast cells were cultured in YPD medium at 25°C for 15 hr to achieve $OD_{600}$ = 0.5. The cells were collected and suspended in the same medium to achieve $OD_{600}$ = 20. 20 mM FM4-64 (*N*-(3-triethylammoniumpropyl)-4-(6-(4-(diethylamino) phenyl) hexatrienyl) pyridinium) dissolved in DMSO (dimethyl sulfoxide) was added (a final concentration of 20 μM) while shielded from light. The cells were incubated with FM4-64 for 15 min at 25°C and then the cells were washed twice with YPD medium. Cells were suspended to achieve $OD_{600}$ = 10 in the same YPD medium or YPD containing reagents (such as rapamycin or PHS), and incubated at 25°C for 2 hr under light shielding to label the vacuoles. After incubation, cells were collected and observed with a fluorescence microscope.

## Western blotting

To analyze Sch9 phosphorylation, protein extracts from cells expressing SCH9-5HA were treated with 2-nitro-5-thiocyanobenzoic acid overnight, resolved by sodium dodecyl sulfate–polyacrylamide gel electrophoresis (SDS–PAGE), immunoblotted, and visualized using rat anti-HA monoclonal antibody (12158167001; Roche) and anti-rat IgG antibody (A9037; Sigma-Aldrich) produced in goat. Bands were quantified using ImageJ to determine the relative amounts of phosphorylated Sch9.

## Coimmunoprecipitation

Coimmunoprecipitation experiment was performed as described in *Rodriguez-Gallardo et al., 2020* and *Rodriguez-Gallardo et al., 2022*. The ER-enriched fraction was solubilized by treatment with 1% digitonin for 1 hr at 4°C, treated with blocked agarose beads (chromotek) then immunoprecipitated with GFP-Trap agarose beads (chromotek). The immunoprecipitated GFP protein complexes were separated by SDS–PAGE and analyzed by immunoblot using rat anti-HA antibody (12158167001; Roche), anti-rat IgG antibody (A9037; Sigma-Aldrich), mouse anti-GFP antibody (11814460001, Roche), and anti-mouse IgG (A4416; Sigma-Aldrich).

## Lipid labeling with [$^3$H]DHS

In vivo labeling of lipids with [$^3$H]DHS was carried out as described (*Ikeda et al., 2021*). Radiolabeled lipids were extracted with chloroform–methanol–water (10:10:3, vol/vol/vol), and analyzed by thin-layer chromatography using a solvent system (chloroform-methanol-4.2N ammonium hydroxide (9:7:2, vol/vol/vol)). Radiolabeled lipids were visualized and quantified on an FLA-7000 system.

## Protein complex modeling

The modeled Tcb1p, Tcb2p, and Tcb3p structures were obtained from AlphaFold2 Protein Structure Database (https://alphafold.ebi.ac.uk/). Based on the modeled structures, the amino acids of TM regions were predicted as 79–171 for Tcb1p, 79–162 for Tcb2p, and 189–268 for Tcb3p. The protein complex structures of the TM regions in Tcb1p, Tcb2p, and Tcb3p were modeled by AlphaFold2 program version 2.3.2. (*Jumper et al., 2021*) worked on the AlphaFold Colab web space (*Mirdita et al., 2022*; https://colab.research.google.com/github/deepmind/alphafold/blob/main/notebooks/AlphaFold.ipynb#scrollTo=pc5-mbsX9PZC). The figures were drawn using PyMOL software provided by Schrödinger, Inc (https://pymol.org/2/).

## Data and materials availability statement

All data that supporting the findings of this study are included in the article as source data files. The materials used in this study are listed in supplementary files and are available from the corresponding author.

## Acknowledgements

We thank Takashi Toda and Masashi Yukawa for GBP-plasmid, Mike Henne for the Δ*NVJ* strain, and Howard Riezman for GFP-*CPS1* plasmid. This research was supported by JSPS KAKENHI 21K20572 and 22K14863 to AI, 19H02922 and 21K19088 to KF.

## Additional information

### Funding

| Funder | Grant reference number | Author |
|---|---|---|
| Japan Society for the Promotion of Science | 21K20572 | Atsuko Ikeda |
| Japan Society for the Promotion of Science | 22K14863 | Atsuko Ikeda |
| Japan Society for the Promotion of Science | 19H02922 | Kouichi Funato |

| Funder | Grant reference number | Author |
|---|---|---|
| Japan Society for the Promotion of Science | 21K19088 | Kouichi Funato |

The funders had no role in study design, data collection, and interpretation, or the decision to submit the work for publication.

## Author contributions

Kazuki Hanaoka, Data curation, Formal analysis, Validation, Investigation; Kensuke Nishikawa, Data curation, Formal analysis, Investigation; Atsuko Ikeda, Data curation, Formal analysis, Validation, Investigation, Writing – original draft, Writing – review and editing; Philipp Schlarmann, Writing – review and editing; Saku Sasaki, Sayumi Yamashita, Aya Nakaji, Data curation, Investigation; Sotaro Fujii, Data curation, Formal analysis, Visualization; Kouichi Funato, Conceptualization, Supervision, Writing – original draft, Project administration, Writing – review and editing

## Author ORCIDs

Atsuko Ikeda ⓘ https://orcid.org/0000-0002-4710-5112
Kouichi Funato ⓘ http://orcid.org/0000-0002-1486-1296

Reviewer #1 (Public Review): https://doi.org/10.7554/eLife.89938.4.sa1
Reviewer #2 (Public Review): https://doi.org/10.7554/eLife.89938.4.sa2
Reviewer #3 (Public Review): https://doi.org/10.7554/eLife.89938.4.sa3
Author Response https://doi.org/10.7554/eLife.89938.4.sa4

# Additional files

## Supplementary files

- Supplementary file 1. Yeast strains used in this study. Related to all figures.
- Supplementary file 2. Plasmids used in this study. Related to all figures.
- MDAR checklist

## Data availability

All data supporting the findings of this study are included in the article as source data files.

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
