## [Editor Report · eLife assessment]

This manuscript presents **valuable** findings that contribute to our understanding of how sphingolipids and membrane contact sites, formed by the tethering protein family tricalbins, are involved in regulating vacuolar morphology in *S. cerevisiae*. The evidence supporting the authors' claims is largely **solid**. While the reported correlation between sphingolipid levels and vacuole homeostasis is interesting and intriguing, more work is needed to thoroughly substantiate the proposed mechanism. This study will be of interest to cell biologists focusing on intracellular organization and lipid metabolism.

---

## [Referee Report · Reviewer #1 (Public Review)]

The manuscript investigates the role of membrane contact sites (MCSs) and sphingolipid metabolism in regulating vacuolar morphology in the yeast *Saccharomyces cerevisiae*. The authors show that tricalbin (1-3) deletion leads to vacuolar fragmentation and the accumulation of the sphingolipid phytosphingosine (PHS). They propose that PHS triggers vacuole division through MCSs and the nuclear-vacuolar junction (NVJ). The study presents some solid data and proposes potential mechanisms underlying vacuolar fragmentation driven by this pathway. Although the manuscript is clear in what the data indicates and what is more hypothetical, the story would benefit from providing more conclusive evidence to support these hypothesis. Overall, the study provides valuable insights into the connection between MCSs, lipid metabolism, and vacuole dynamics.

---

## [Referee Report · Reviewer #2 (Public Review)]

This manuscript explores the mechanism underlying the accumulation of phytosphingosine (PHS) and its role in initiating vacuole fission. The study posits the involvement of membrane contact sites (MCSs) in two key stages of this process. Firstly, MCSs tethered by tricalbin between the endoplasmic reticulum (ER) and the plasma membrane (PM) or Golgi regulate the intracellular levels of PHS. Secondly, the amassed PHS triggers vacuole fission, most likely through the nuclear-vacuolar junction (NVJ). The authors propose that MCSs play a regulatory role in vacuole morphology via sphingolipid metabolism.

While some results in the manuscript are intriguing, certain broad conclusions occasionally surpass the available data. Despite the authors' efforts to enhance the manuscript, certain aspects remain unclear. It is still uncertain whether subtle changes in PHS levels could induce such effects on vacuolar fission. Additionally, it is regrettable that the lipid measurements are not comparable with previous studies by the authors. Future advancements in methods for determining intracellular lipid transport and levels are anticipated to shed light on the remaining uncertainties in this study.

---

## [Referee Report · Reviewer #3 (Public Review)]

In this manuscript, the authors investigated the effects of deletion of the ER-plasma membrane/Golgi tethering proteins tricalbins (Tcb1-3) on vacuolar morphology to demonstrate the role of membrane contact sites (MCSs) in regulating vacuolar morphology in *Saccharomyces cerevisiae*. Their data show that tricalbin deletion causes vacuolar fragmentation possibly in parallel with TORC1 pathway. In addition, their data reveal that levels of various lipids including ceramides, long-chain base (LCB)-1P, and phytosphingosine (PHS) are increased in tricalbin-deleted cells. The authors find that exogenously added PHS can induce vacuole fragmentation and by performing analyses of genes involved in sphingolipid metabolism, they conclude that vacuolar fragmentation in tricalbin-deleted cells is due to the accumulated PHS in these cells. Importantly, exogenous PHS- or tricalbin deletion-induced vacuole fragmentation was suppressed by loss of the nucleus vacuole junction (NVJ), suggesting the possibility that PHS transported from the ER to vacuoles via the NVJ triggers vacuole fission. Of note, the authors find that hyperosmotic shock increases intracellular PHS levels, suggesting a general role of PHS in vacuole fission in response to physiological vacuolar division-inducing stimuli.

This work provides valuable insights into the relationship between MCS-mediated sphingolipid metabolism and vacuole morphology. The conclusions of this paper are mostly supported by their results, but inclusion of direct evidence indicating increased transport of PHS from the ER to vacuoles via NVJ in response to vacuolar division-inducing stimuli would have strengthened this study.

There is another weakness in their claim that the transmembrane domain of Tcb3 contributes to the formation of the tricalbin complex which is sufficient for tethering ER to the plasma membrane and the Golgi complex. Their claim is based only on the structural simulation, but not on by biochemical experiments such as co-immunoprecipitation and pull-down.

---

## [Author Response]

The following is the authors’ response to the previous reviews.

**Reviewer #1 (Public Review):**
The manuscript investigates the role of membrane contact sites (MCSs) and sphingolipid metabolism in regulating vacuolar morphology in the yeast *Saccharomyces cerevisiae*. The authors show that tricalbin (1-3) deletion leads to vacuolar fragmentation and the accumulation of the sphingolipid phytosphingosine (PHS). They propose that PHS triggers vacuole division through MCSs and the nuclear-vacuolar junction (NVJ). The study presents some solid data and proposes potential mechanisms underlying vacuolar fragmentation driven by this pathway. Although the manuscript is clear in what the data indicates and what is more hypothetical, the story would benefit from providing more conclusive evidence to support these hypothesis. Overall, the study provides valuable insights into the connection between MCSs, lipid metabolism, and vacuole dynamics.

We thank the positive review from the Reviewer #1. We hope that our hypotheses are supported by the "Author Response to Recommendations" and by further research in the future.

**Reviewer #2 (Public Review):**
This manuscript explores the mechanism underlying the accumulation of phytosphingosine (PHS) and its role in initiating vacuole fission. The study posits the involvement of membrane contact sites (MCSs) in two key stages of this process. Firstly, MCSs tethered by tricalbin between the endoplasmic reticulum (ER) and the plasma membrane (PM) or Golgi regulate the intracellular levels of PHS. Secondly, the amassed PHS triggers vacuole fission, most likely through the nuclear-vacuolar junction (NVJ). The authors propose that MCSs play a regulatory role in vacuole morphology via sphingolipid metabolism. While some results in the manuscript are intriguing, certain broad conclusions occasionally surpass the available data. Despite the authors' efforts to enhance the manuscript, certain aspects remain unclear. It is still uncertain whether subtle changes in PHS levels could induce such effects on vacuolar fission. Additionally, it is regrettable that the lipid measurements are not comparable with previous studies by the authors. Future advancements in methods for determining intracellular lipid transport and levels are anticipated to shed light on the remaining uncertainties in this study.

We thank the careful comment from Reviewer #2. As Reviewer #2 pointed out, the mechanism of how slight changes in PHS levels can induce the vacuolar fission event is still uncovered in this manuscript. We sincerely consider that this issue has to be resolved in further study.

**Reviewer #3 (Public Review):**
In this manuscript, the authors investigated the effects of deletion of the ER-plasma membrane/Golgi tethering proteins tricalbins (Tcb1-3) on vacuolar morphology to demonstrate the role of membrane contact sites (MCSs) in regulating vacuolar morphology in *Saccharomyces cerevisiae*. Their data show that tricalbin deletion causes vacuolar fragmentation possibly in parallel with TORC1 pathway. In addition, their data reveal that levels of various lipids including ceramides, long-chain base (LCB)-1P, and phytosphingosine (PHS) are increased in tricalbin-deleted cells. The authors find that exogenously added PHS can induce vacuole fragmentation and by performing analyses of genes involved in sphingolipid metabolism, they conclude that vacuolar fragmentation in tricalbin-deleted cells is due to the accumulated PHS in these cells. Importantly, exogenous PHS- or tricalbin deletion-induced vacuole fragmentation was suppressed by loss of the nucleus vacuole junction (NVJ), suggesting the possibility that PHS transported from the ER to vacuoles via the NVJ triggers vacuole fission. Of note, the authors find that hyperosmotic shock increases intracellular PHS levels, suggesting a general role of PHS in vacuole fission in response to physiological vacuolar division-inducing stimuli. This work provides valuable insights into the relationship between MCS-mediated sphingolipid metabolism and vacuole morphology. The conclusions of this paper are mostly supported by their results, but inclusion of direct evidence indicating increased transport of PHS from the ER to vacuoles via NVJ in response to vacuolar division-inducing stimuli would have strengthened this study. There is another weakness in their claim that the transmembrane domain of Tcb3 contributes to the formation of the tricalbin complex which is sufficient for tethering ER to the plasma membrane and the Golgi complex. Their claim is based only on the structural simulation, but not on by biochemical experiments such as co-immunoprecipitation and pull-down.

We appreciate the careful feedback from Reviewer #3. We have responded in the "Recommendations to Authors" section and hope it can partially support the weakness in our claim regarding the physical interaction between Tcb1, 2, and 3.

**Reviewer #1 (Recommendations For The Authors):**
I would suggest that the authors include some of the data (e.g., Tcb interactions) that they refer to in the response to the reviewers. I think that this could enhance the message in this manuscript. Also, maybe it's a typo and you were referring to some other image panel, but in the rebuttal letter a "Fig. S3B" is mentioned, but I could not find it.

Following the suggestions of reviewers #1 and #3, we have added the data of co-immunoprecipitation which confirmed that Tcb3 binds to both Tcb1 and Tcb2 as Supplemental Figure 2. With this change, the person (Ms. Saku Sasaki) who performed this analysis was also added as a co-author.

Also, we appreciate the careful remark and apologize for the mistake. In the previous Author's response, we mentioned the vacuole observation using SD medium, but this data was Fig 5C, not Fig S3B.

**Reviewer #3 (Recommendations For The Authors):**
I would recommend that the authors include the IP data mentioned in their rebuttal letter to show the interactions among Tcb1-3. Also, the authors should quantify all lipid species in Fig 5B, as shown in Fig 3A.

Following the suggestions of reviewers #1 and #3, we have added the co-immunoprecipitation data (Fig S2). In a further study, we would like to test if the transmembrane domain of Tcb3 is sufficient for the interaction among Tcb1-3. Also, we quantified all lipid species and replaced the data in Fig 5B.

Minor points:(1) The function of vps4 is not mentioned in the manuscript.(2) The function of Sur2p is not mentioned in the manuscript. It should be clearly mentioned that DHS is converted to PHS by Sur2p.

(1) We have added text sections which mention that VPS4 is needed for normal ESCRT function, and its deletion is an example for inhibition of GFP-Cps1p transport into the vacuole.

(2) We have added the text in the manuscript that states Sur2p is the hydroxylase that catalysis the conversion of DHS to PHS.